# Knowledge, Attitude and Practice Regarding Antibacterial and Their Resistance in Medellín-Colombia: A Cross-Sectional Study

**DOI:** 10.3390/antibiotics12071101

**Published:** 2023-06-25

**Authors:** Marcela Sampedro Restrepo, Manuela González Gaviria, Samuel Arango Bolaños, Luis Felipe Higuita-Gutiérrez

**Affiliations:** 1School of Microbiology, Universidad de Antioquia, Medellín 050010, Colombia; marcela.sampedro@udea.edu.co (M.S.R.); manuela.gonzalezg@udea.edu.co (M.G.G.); samuel.arangob@udea.edu.co (S.A.B.); 2School of Medicine, Universidad Cooperativa de Colombia, Medellín 050012, Colombia

**Keywords:** drug resistance, antibiotics, health education, KAP model, questionnaire

## Abstract

Objective: To describe the knowledge, attitudes, and practices (KAPs) profile on bacterial resistance and antibiotic in the general population of Medellín. Material and methods: A cross-sectional study was conducted from September to December 2022, with 902 participants selected through stratified sampling with proportional allocation of different sectors of the city. The KAP scale was developed through a literature review, elimination of duplicate items, validity assessment, and reliability evaluation using Cronbach’s alpha. Each item was presented with absolute and relative frequencies on a Likert scale, with a total score ranging from 0 to 100, where a higher score indicates better knowledge, attitudes, and practices. Comparisons were made using Mann-Whitney U, Kruskal-Wallis H, and linear regression. Results: The knowledge score median was 73.3 (IQR 63.3–93.3), with 36.9% reporting that antibiotics can be stopped once symptoms improve and 26.1% considering them as analgesics or antipyretics. The attitudes score was 83.3 (IQR 73.3–93.3), with 95.3% expressing concern about the impact on their health or that of their family and over 90% agreeing that more information is needed on antibiotic resistance. The practice score was the lowest at 63.9 (IQR 50–75), with 48% having been prescribed antibiotics at the pharmacy and 42.6% taking them to treat flu symptoms. Economic status (β 2.645), education in health-related areas (β 6.224), gender (β 2.892), and education level (β 3.257) determined knowledge. Knowledge (β 0.387), gender (β 2.807), and education level (β 0.686) influenced attitudes, but practices were only determined by Knowledge (β 0.084) attitudes (β 0.552) and age group (β 2.858). Conclusions: Knowledge about antibiotics and bacterial resistance does not significantly influence the practices of the population. Therefore, interventions aimed at improving knowledge need to be reconsidered as they may not contribute to the appropriate use of antibiotics and prevention of resistance to these drugs.

## 1. Introduction

Bacterial resistance has emerged as a significant global concern. The World Health Organization (WHO) regards it as one of the top 10 health threats [1]. According to Murray et al. [2], antimicrobial resistance (AMR) caused 4.95 million deaths in 2019, with 1.27 million potentially preventable. AMR has affected not only health but also the economy [3]. The World Bank has projected that up to 3.8% of the global gross domestic product could be lost due to AMR by 2050 [3].

Recognizing the magnitude of this problem, the WHO established the Global Antimicrobial Resistance Surveillance System (GLASS) in 2015. The 2022 report by GLASS indicates that oral antibiotics are the primary route of antibiotic consumption, with most antibiotic use occurring at the community level [3]. The general population plays a critical role in bacterial resistance due to inadequate practices such as self-medication, interruption of prescriptions, lack of therapeutic adherence, limited knowledge, and poor attitudes toward prevention [4,5]. To address this issue, the United Nations General Assembly on Antimicrobial Resistance committed to promoting activities that increase the general population’s knowledge of antimicrobial resistance to encourage changes in their attitudes and practices towards these drugs [6]. A greater understanding of the topic is assumed to lead to better attitudes and practices.

Given the paramount importance of knowledge, attitudes, and practices (KAP) regarding the use of antibiotics and bacterial resistance, several studies have been conducted in diverse regions worldwide as China [7], Saudi Arabia [8], South Africa [9], Ethiopia [10], Germany [11], Romania [12], Canada [13], and Peru [14]. These studies demonstrate significant heterogeneity in the proportion of the population that holds certain beliefs and engages in certain antibiotic-related behaviours. These variations include beliefs about the appropriateness of using antibiotics for the common cold, switching antibiotics during treatment, self-medication with antibiotics, and abruptly discontinuing treatment. Furthermore, previous studies do not describe the measurement process for the knowledge, attitudes, and practices (KAP) indicators, the boundaries between each index, the construction process, or the reliability properties, internal consistency, and discriminant power of the instruments used.

Specifically in Colombia, the national government has developed strategies for the detection and control of this problem. Among the strategic lines it defined was to provide information to the general population about the safe use of antimicrobials [15]; however, studies on KAP regarding antibiotics in the general population are scarce. In Medellin, the second most important city in the country, a study conducted on 532 medical students revealed that 69.3% knew that self-medication contributes to resistance, 11.2% believed that antibiotics should be stopped when symptoms disappear, and 28.5% preferred not to act on the problem of resistance because they believed that their actions would have a little impact [16]. In this study and the fact that antibiotics are sold without a prescription in the city [17], many antibiotic-resistant bacteria are reported. New resistance mechanisms frequently appear [18], suggesting that KAP in the general population is deficient. Therefore, this study was designed to describe the profile of knowledge, attitudes, and practices regarding bacterial resistance and the use of antibiotics in the general population of Medellin, Colombia.

## 2. Materials and Methods

### 2.1. Study Design

A cross-sectional descriptive study was conducted, comprising 900 participants who resided in Medellin and voluntarily agreed to participate between August and December of 2022.

### 2.2. Population, Sample, and Sampling

The sample size was calculated based on a reference population of 2,000,000 individuals over 18 years of age in the city [19]. An expected standard deviation of 12 points in the score of each index, a 95% confidence level, a precision of 1, a design effect of 1.5, and a sampling correction of 10% were also considered. Therefore, the study aimed to include 914 participants. A design effect greater than 1 was assumed because the city is divided into six zones and rural districts. The sample sought to maintain the proportion of individuals contributed by each zone to the total population. The participants were selected using stratified proportional allocation sampling within each zone or rural district. They were approached at the busiest points of each sector, such as parks, shopping centers, sports facilities, and areas near metro stations (the city’s main mass transportation system). The final sampling consisted of the following: rural districts (*n* = 113), northeastern zone (*n* = 158), northwestern zone (*n* = 179), eastern-central zone (*n* = 191), western-central zone (*n* = 126), southeastern zone (*n* = 39), and southwestern zone (*n* = 96).

### 2.3. Survey 

The scale was constructed in 3 stages. Firstly, a literature review was conducted in various databases such as Pubmed, Scielo, and Google Scholar using the terms “antibiotic” or “antimicrobial” in conjunction with “knowledge” or “belief” and “survey” or “questionnaire.” Some of the algorithms used were: ((Antibiotic[Title/Abstract]) AND (Survey[Title/Abstract])) AND (belief[Title/Abstract]); ((Antimicrobial[Title/Abstract]) AND (survey[Title/Abstract])) AND (knowledge[Title/Abstract]); ((Antimicrobial[Title/Abstract]) AND (questionnaire[Title/Abstract])) AND (belief[Title/Abstract]). The search identified 2600 articles, of which 1002 duplicates were eliminated. After screening for the inclusion criteria, 312 articles were obtained, and 242 items were extracted. From these, questions that evaluated KAPs on a quantifiable scale and established clear boundaries between knowledge, attitudes, and practices were chosen. A pilot test was conducted on 30 people to verify that the questions were clear, well-worded, and contributed to the project’s objectives. The final questionnaire was developed in Spanish and covered four sections: the first provided sociodemographic information (with a total of 5 items), the second included questions about knowledge (10 items), the third about attitudes (10 items), and the fourth with practices (12 items) related to antibiotics and bacterial resistance. Each index (knowledge, attitudes, and practices) was evaluated with Likert scale items of 4 levels, with 1. Completely disagree, 2. Disagree, 3. Agree, or 4. Completely agree with knowledge and attitudes and for practices as 1. Never, 2. Almost never, 3. Almost always, and 4. Always. Each index generated a score from 0 (worst possible score) to 100 (best possible score) using the following formula: Total score = [(obtained score—lowest possible score)/maximum possible score—minimum possible score] × 100. The reliability (Cronbach’s alpha), Internal consistency, discriminating power, ceiling effect and floor effect were calculated.

### 2.4. Data Analysis 

For descriptive analysis, the gathered data was examined using frequency tables, relative frequencies, and summary measures such as position, dispersion, and central tendency. Given the non-normal distribution of the data, as determined by the Kolmogorov-Smirnov test with Lilliefors correction, we presented the total score as the median and interquartile range. To compare knowledge, attitudes, and practices across different variables such as age group, gender, economic status, educational level, and training in the health area, we employed the Mann-Whitney U test and Kruskal-Wallis H test. Finally, we conducted linear regressions to identify potential confounding variables for the associations found in the bivariate analysis. The regression models underwent several assumption validation tests. Firstly, the randomness of the score of each dimension was assessed using the run test. Secondly, linearity was examined using ANOVA. The normality and constant variance of the residuals were evaluated, while the uncorrelation of residuals was concerned with the Durbin-Watson test. Lastly, the non-multicollinearity assumption was assessed by estimating the tolerance of each variable and the inflation factor of the variance VIF.

Reliability: The instrument’s reliability was assessed using Cronbach’s alpha, with values greater than 0.7 considered satisfactory.

Internal consistency: The internal consistency of the instrument was evaluated using the Spearman correlation coefficient between the items and the respective index they belong to. Values equal to or greater than 0.4 were considered satisfactory. The success percentage was calculated using the following formula:Success percentage = (# of item-index correlations to which it belongs ≥ 0.4)/(total correlations of each index) × 100

Discriminating power: To assess the discriminating power of the instrument, Spearman correlations were computed between items and the index they do not belong to. Values lower than the correlation of the items with their respective indices were considered satisfactory. The success percentage was calculated using the formula:Success percentage = (# of item-index correlations to which it does not belong < item-index correlations to which it belongs)/(total number of item-index correlations to which it does not belong) × 100

Floor effect: was determined by calculating the percentage of participants who obtained the minimum possible score on each index.

Ceiling effect: was determined by calculating the percentage of participants who obtained the maximum possible score on each index.

All statistical procedures were performed using SPSS version 27.0, with a significance level set at *p* < 0.05.

## 3. Results

The survey included 902 participants whose mean age ± standard deviation was 44.8 ± 17.3 years, with an age range of 18 to 90 years. Among the respondents, 52.6% (95% CI 49.3–55.9) were female, 2.0% (95% CI 1.2–3.1) had no formal education, and 12.3% had educational qualifications in the health field, as shown in Table 1.

### 3.1. Knowledge Index

The median score for the knowledge index was 73.3 (IQR 63.3–93.3), and it was found that 88.8% of the participants were aware of the danger of antibiotic resistance for human health, and 84.4% acknowledged the significant problem of resistance in the country. However, 36.9% believed that antibiotic treatment could be stopped when symptoms improve, 31.3% thought that taking antibiotics without a prescription does not contribute to bacterial resistance, 29.1% were unaware of the possibility of infections due to bacteria resistant to all available antibiotics, and 26.1% considered antibiotics as analgesics or antipyretics (as presented in Figure 1). 

### 3.2. Attitude Index

The median attitude score was 83.3 (IQR 73.3–93.3). The index revealed that 95.3% of respondents expressed concern regarding the potential impact of bacterial resistance on their health or that of their family. Additionally, over 90% of respondents agreed that more information regarding antibiotics and bacterial resistance is required. However, 36.3% of respondents expect doctors to prescribe antibiotics for common cold symptoms, and 22.7% consider doctors who do not prescribe antibiotics when requested as bad doctors (Figure 2). 

### 3.3. Practice Index

The median practice score was the lowest of the three indices, with 63.9 (IQR 50–75) points, and 12.5% of respondents reported having taken measures to protect family and friends from bacterial resistance. On the other hand, 58.9% reported having taken leftover antibiotics from a previous illness, 48% have been recommended to take antibiotics at the pharmacy, 42.6% have taken antibiotics for the treatment of flu symptoms, and 16.4% have experienced antibiotic treatment was not effective (Figure 3). 

### 3.4. Factors Associated with Knowledge, Attitude and Practice

The bivariate analysis revealed significant associations between age group, gender, economic status, educational level, and training in health-related areas with knowledge, attitudes, and practice scores. The median scores can be found in Table 2, while the multiple comparisons, adjusted with the Bonferroni test, are provided in the Appendix A.

### 3.5. Relationship between Knowledge, Attitudes, and Practices

To validate whether the theoretical model matched the population’s behavior, we performed linear regression models for each index. We discovered that economic status, education in health-related areas, gender, and education level impact knowledge. In turn, knowledge, gender, and education levels influence attitudes. However, practices are solely influenced by attitudes and age groups. The relationship between knowledge and practices was almost negligible, with a beta coefficient of 0.084 (Table 3).

### 3.6. Psychometric Properties of the Instrument

Psychometric properties of the KAP scale were assessed, and the results indicate excellent reliability, internal consistency, and discriminant power (Appendix A).

## 4. Discussion

Recently, various initiatives have been undertaken to enhance public awareness about antibiotics and bacterial resistance to reduce self-medication and inappropriate use of these drugs [6]. The aim is to prevent what Margaret Chan, the former Director-General of the World Health Organization, called the post-antibiotic era [20]. This study, conducted among the general population of Medellín, Colombia, revealed that people’s knowledge of this topic does not necessarily influence their practices.

Regarding the knowledge index, the study found that the majority of the population recognizes the threat posed by antibiotic resistance to health, understands that resistance not only affects hospitalized individuals, and acknowledges that consuming antibiotics without a prescription contributes to bacterial resistance. These findings align with other studies conducted worldwide. For example, in Singapore, 78.2% of respondents were aware that the problem of resistance not only affects hospitalized people [21]. In the Chinese population, 80% were familiar with the dangers posed by bacterial resistance to human health [7]. Despite the recommendation of the World Health Organization (WHO) [22] and the national plan for addressing bacterial resistance, the city lacks systematic government campaigns to enhance the population’s knowledge on the subject. Therefore, it has also been described that in cases where centralized information about a disease is not provided, knowledge can arise through first-hand observation and word of mouth [23]. 

Knowledge about antibiotics in the city could be improved by focusing on issues such as adherence to antibiotic treatment, the risks of self-medication, and the possibility of infections by multi-resistant bacteria. It is crucial to design actions in this regard because feeling better or improving symptoms does not always indicate that the infection has completely disappeared. Recent studies suggest that shorter antibiotics can be as effective or even better than longer cycles for some infections, reducing adverse effects and bacterial resistance [24]. However, the optimal duration of therapy should be discussed with a treating physician [24]. Moreover, previous studies have described that people aware of a disease’s presence in their environment can reduce their susceptibility to it, potentially stopping its spread altogether [23]. Therefore, there is a need for campaigns specifically to raise awareness about the risk of infections caused by multidrug-resistant bacteria in the city.

Attitudes toward the use of antibiotics and bacterial resistance were generally positive. The study revealed that the community should receive more information on this topic. Concerns exist about the impact that bacterial resistance could have on the health of individuals and their families, and many people believe that they should advise their close contacts not to take antibiotics without a doctor’s prescription. Similar results were found in studies conducted in Europe, where 76.4% of respondents were concerned about the impact of resistance on their health and that of their families [25], and in Australia, where 35% of respondents felt that they should know more about antibiotics and resistance [26], and in the United States where over 90% believed that antibiotics should be used more responsibly [27]. The fact that the population demands more information on this topic represents an opportunity for local authorities to design interventions to improve antibiotic use. A recent systematic literature review identified 43 interventions to change behavior to reduce inappropriate antibiotic use in low- and middle-income countries [28]. Therefore, local authorities could learn from these experiences and design similar interventions in the city.

The practices within this group exhibited room for improvement. Most of the population reported taking leftover antibiotics from previous illnesses, obtaining antibiotics quickly from pharmacies without a prescription, and a considerable proportion mentioned using antibiotics to treat the common cold. It is important to note that practices were influenced solely by attitudes and age groups. These practices can be compared with similar studies conducted in Cambodia, where 85% hoard leftover antibiotics for future use [29], in Peru, where 67.1% also easily obtain antibiotics without a prescription [14], and in Romania, where around 40% of people have taken antibiotics for the flu or common cold [12]. The excessive use of antibiotics in humans is a global problem and proportionally contributes the most to bacterial resistance with the most robust evidence [30]. A significant portion of actions aimed at containing the problem has focused on improving the knowledge of the population, assuming that better knowledge leads to better practices. For example, a systematic review of the literature on public health interventions on this topic found that the 17 identified articles showed some impact on knowledge about antibiotics and bacterial resistance, but none measured the long-term impact of using these drugs [31]. Focusing interventions on improving knowledge on this topic is debatable, as our study revealed that knowledge did not correlate with practices. Several studies have shown that to change practices or behaviors in a population. Health programs must address multiple factors, including sociocultural, environmental, normative, economic, and social determinants in general [32,33].

Another noteworthy finding of this study is the strong psychometric properties exhibited by the scale, demonstrating its high reliability, internal consistency, and discriminant power. It is worth mentioning that a study in the literature has described the development and validation of a questionnaire for assessing KAP regarding personal antibiotic use in Spain [34]. However, our study differs from the previous one as it incorporates elements related to bacterial resistance. It is important to note that our research did not include test-retest reliability or confirmatory factor analysis. Future studies should aim to evaluate these properties.

Additionally, it is important to consider that the Likert scale used in our questionnaire for the knowledge and attitude indices does not include a neutral option. This decision was made for two reasons: first, respondents often tend to opt for easier answer options or simple solutions when faced with questions that require significant cognitive effort, and second, interviewees may choose neutral answers when they are not prepared to disclose socially unacceptable practices or attitudes [35]. Researchers planning to utilize this scale should bear in mind that psychometric properties are not inherent to the scale itself. Therefore, questionnaires validated in a specific country require testing and adaptation before being effectively used in another country with different language and cultural contexts [36,37].

One of the strengths of this study is that it is the first to be carried out in a robust sample size of the Colombian population, providing essential insights into KAP (Knowledge, Attitudes, and Practices) regarding antibiotics and antimicrobial resistance. This study also provides a baseline for future research. It has resulted in the construction of a reliable instrument that could be used in other cities in Colombia or other countries. However, some weaknesses should be highlighted. The questionnaire was developed through a deductive construction process, where the researchers defined the items based on their relevance as described in the literature. Future studies could adopt an inductive approach, using qualitative research methods to refine and adjust the questionnaire. Another limitation of this study is its cross-sectional design, as knowledge, attitudes, and practices are subject to change over time due to personal experiences and interactions with healthcare professionals. Therefore, caution must be exercised when generalizing these findings to populations that differ significantly from the one studied, and it is recommended to validate the questionnaire in such populations before extrapolating the results.

Furthermore, it is worth noting that certain statements in the questionnaire encompass multiple topics, making it challenging for respondents to address specific aspects. Consequently, rephrasing these statements to clarify the specific focus of respondents would be beneficial for future investigations. Finally, the outcome variable in regression models is ideally not restricted by boundaries. However, in this study, the scale generated a score ranging from 0 to 100. Therefore, it is important to consider the calculation of the floor and ceiling effects when analyzing the data. These effects refer to the possibility of a disproportionate number of responses at the lower or upper end of the scale, respectively. Taking these effects into account can help ensure accurate interpretation and analysis of the results.

## 5. Conclusions

This study characterizes the knowledge, attitudes, and practices related to antibiotics and antimicrobial resistance in Medellín, Colombia. The findings highlight a concerning situation regarding inappropriate practices, aligning with evidence from other countries indicating that the general population and improper use of these drugs play a significant role in driving antibiotic resistance. Contrary to the expectations of the KAP model, which suggests that improving knowledge leads to improved practices, the findings of this study reveal that this relationship does not hold in the population of Medellín. This discrepancy emphasizes the need to reconsider the design of national and international initiatives that promote the rational use of antibiotics in the community. While enhancing knowledge is essential, it may not effectively address bacterial resistance in this population. Additionally, this study offers a valuable resource for researchers interested in this field by providing a scale with excellent psychometric properties, such as reliability, internal consistency, and discriminant power. This scale can be adapted and utilized in future research conducted in other cities within our country or even in different countries, contributing to advancing knowledge on this topic.

## Figures and Tables

**Figure 1 antibiotics-12-01101-f001:**
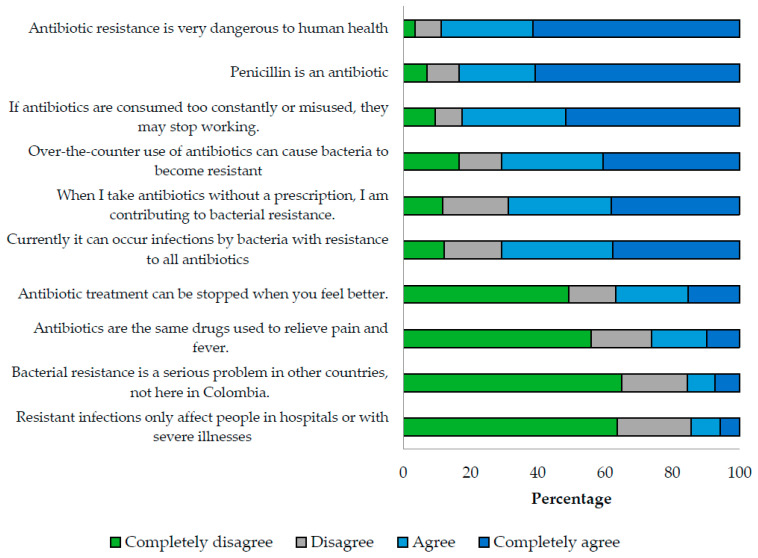
Relative frequencies of knowledge about antibiotics and antibiotic resistance.

**Figure 2 antibiotics-12-01101-f002:**
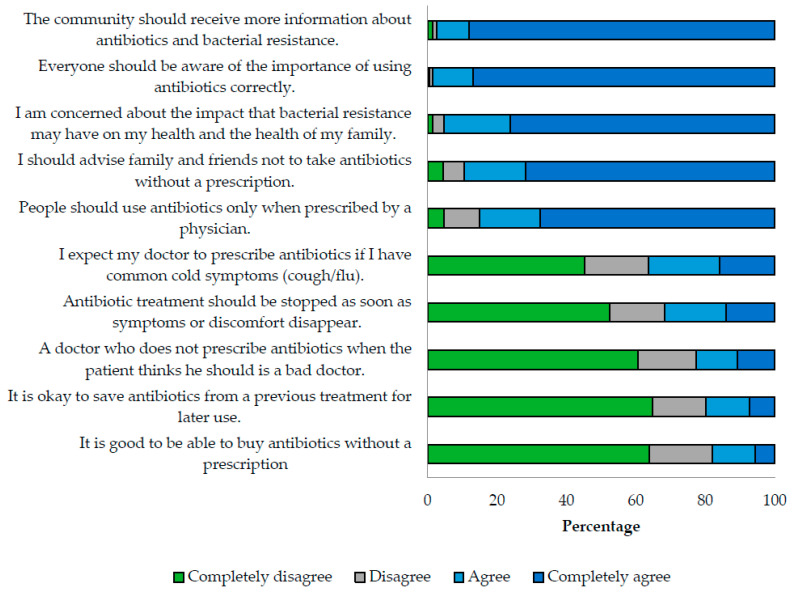
Relative frequencies of attitudes about antibiotics and antibiotic resistance.

**Figure 3 antibiotics-12-01101-f003:**
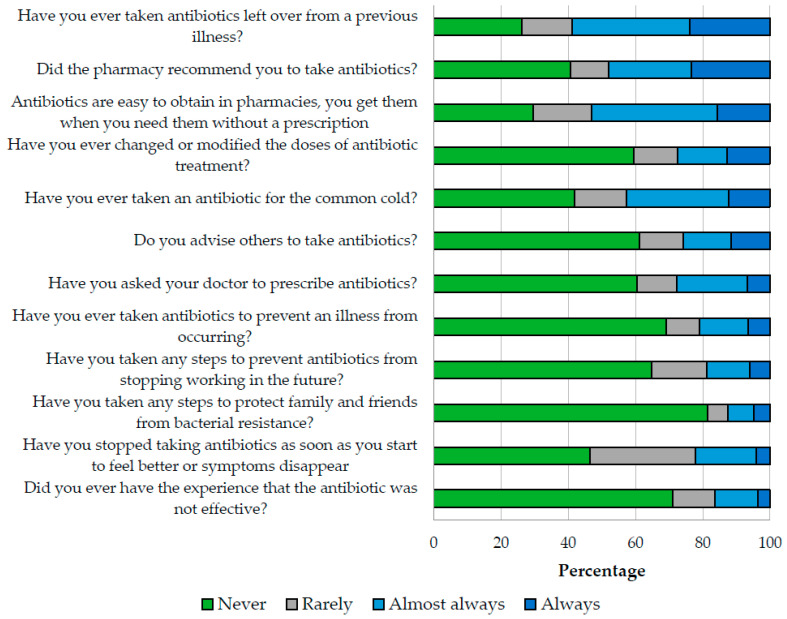
Relative frequencies of practices about antibiotics and antibiotic resistance.

**Table 1 antibiotics-12-01101-t001:** Description of the sociodemographic characteristics of the study participants.

		*n*	% (CI 95%)
Age Group ^ǂ^	Youth	167	18.5 (16.1–21.1)
	Adult	515	57.1 (53.8–60.3)
	Older adult	220	24.4 (21.7–27.3)
Gender	Female	474	52.6 (49.3–55.9)
	Male	427	47.4 (44.1–50.7)
Economic status	Low	414	47.5 (44.2–50.9)
	Medium	432	49.6 (46.3–52.9)
	High	25	2.9 (1.9–4.1)
Education level	None	18	2.0 (1.2–3.1)
	Primary school	199	22.1 (19.4–24.9)
	High school	286	31.7 (28.7–34.8)
	Technical school	160	17.7 (15.4–20.3)
	College student	107	11.9 (9.9–14.1)
	Professional	132	14.6 (12.4–17.1)
Healthcare career	No	781	87.7 (85.4–89.7)
	Yes	110	12.3 (10.3–14.6)

Note: In the question about education in areas related to health and economic status, the table shows figures lower than 902 due to missing data. ^ǂ^ Age was categorized into groups according to the Ministry of Health’s classification of life cycles: 18–26 years = youth; 27–59 years = adulthood, and 60 years or older = old age. The mean and standard deviation in the young group was 22 ± 3. In the adults, it was 42 ± 10, and in the older adults, it was 68 ± 6.

**Table 2 antibiotics-12-01101-t002:** Comparison of knowledge, attitudes, and practices according to sociodemographic characteristics.

	Knowledge	Attitudes	Practices
	Me (IQR)	Me (IQR)	Me (IQR)
Age group			
Youth	73.3 (56.7–86.7)	83.3 (70.0–93.3)	61.1 (47.2–72.2)
Adult	76.7 (63.3–86.7)	83.3 (73.3–93.3)	61.1 (50.0–75.0)
Older adult	70.0 (60.0–80.0)	80.0 (70.0–90.0)	66.7 (52.8–80.6)
*p*-value ^ʄ^	0.001 *	0.021 *	0.006 *
Gender			
Female	76.7 (63.3–86.7)	86.7 (76.7–93.3)	63.9 (50.0–77.8)
Male	73.3 (60.0–83.3)	80.0 (70.0–90.0)	61.1 (47.2–75.0)
*p*-value ^ʃ^	0.005 *	0.001 *	0.018 *
Economic status			
Low	70.0 (60.0–80.0)	83.3 (70.0–90.0)	61.1 (47.2–75.0)
Medium	76.7 (65.0–90.0)	86.7 (73.3–93.3)	63.9 (52.8–77.8)
High	80.0 (70.0–90.0)	86.7 (76.7–93.3)	61.1 (58.3–69.4)
*p*-value ^ʄ^	0.001 *	0.003 *	0.005 *
Education level			
None	61.7 (56.7–70.0)	71.7 (63.3–80.0)	58.3 (47.2–69.4)
Primary school	66.7 (56.7–76.7)	80.0 (70.0–86.7)	63.9 (52.8–75.0)
High school	70.0 (60.0–80.0)	83.3 (70.0–90.0)	58.3 (47.2–72.2)
Technical school	76.7 (70.0–86.7)	86.7 (76.7–93.3)	61.1 (52.8–75.0)
College student	83.3 (66.7–93.3)	90.0 (76.7–96.7)	66.7 (52.8–80.6)
Professional	83.3 (76.7–93.3)	90.0 (76.7–100)	66.7 (51.4–80.6)
*p*-value ^ʄ^	0.001 *	0.001 *	0.008 *
Healthcare career			
No	70.0 (60.0–83.3)	83.3 (73.3–90.0)	61.1 (50.0–75.0)
Yes	86.7 (73.3–96.7)	90.0 (80.0–96.7)	69.4 (55.6–80.6)
*p*-value ^ʃ^	0.001 *	0.001 *	0.002 *

Note: Me: Median; IQR: Interquartile range; ^ʃ^: Mann-Whitney U test; ^ʄ^: Kruskall-Wallis H test; * This is a statistically significant result

**Table 3 antibiotics-12-01101-t003:** Linear regression results for knowledge, attitudes, and practices.

Index	Variables of the Model	Regression Coefficient	CI 95%	*p*-Value	Coefficient of Determination
Knowledge	Healthcare career (Yes/No)	6.224	3.170–9.279	<0.001	17.2%
	Education level ^ʃ^	3.257	0.993–4.792	<0.001	
	Gender (Female/Male)	2.892	2.498–4.016	0.003	
	Economic status ^ʃ^	2.645	0.837–4.453	0.004	
Attitudes	Knowledge Index	0.387	0.331–0.443	<0.001	23.1%
	Gender (Female/Male)	2.807	1.199–4.415	<0.001	
	Education level ^ʃ^	0.686	0.062–1.309	0.031	
Practices	Knowledge Index	0.084	0.005–0.157	0.036	22.4%
	Attitudes index	0.552	0.468–0.636	<0.001	
	Age group ^ʃ^	2.858	1.273–4.443	<0.001	

^ʃ^ Note: The order in the categories of the variables and the regression coefficient is as follows. Model Knowledge Education level: None = Reference, Primary school = −71.179, High school = −16.835, Technical school = 9.678, College student = 8.963, Professional = 11.288. Economic status: Low = reference, Middle = 3.275, High = 0.204. Model attitudes Education level: None = Reference, Primary school = −14.173, High school = −0.609 Technical school = 1.937, College student = 2.255, Professional = 1.599. Model practices Age group Youth = Reference, Adult = −6.829, Older adult = 10.366.

## Data Availability

Data has not been deposited in a public repository. Anonymised data is available on reasonable request to the authors.

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
