# Peer review of "Knowledge, Attitude and Practice Regarding Antibacterial and Their Resistance in Medellín-Colombia: A Cross-Sectional Study"

_antibiotics, 2023, doi:10.3390/antibiotics12071101_

Round 1

Reviewer 1 Report

Thank you for your work and for the opportunity to review this paper.

Materials and Methods 

After the introduction and before the results, it would be interesting for the authors to include a section on methodology (section 4) specifying the design of the survey, in which population it was applied, and a brief explanation of the scales used on the items they include.

Results

In line 93 it would be convenient for the authors to indicate the measure of dispersion of the mean, for example standard deviation.

It would be interesting if the average age and standard age distinction in the youth, adult and older adult groups were provided.

In the regression model for the variables associated with knowledge, attitudes and practices It would be interesting to provide a table, which included specific statistical data, apart from the specific Beta coefficients and p-values of each factor, it is also interesting to explain the R2 model (coefficient of determination).

Discussion

Limitations: This section is very superficial. Limitations relates to the deployment of the research and consequences are for the team to own, not the participants. Bias is related to interpretation of the researchers. If there was potential for participants to misinterpret, then a limitation was the team not performing a pilot, or testing the questions in a content analysis prior to recruitment.

Limitations should be presented as where concerns exist after the team has reflected, and what could be done to mitigate that concern if the reader chooses to replicate the study. 

Conclusions 

It would be advisable for the authors to explain in the conclusions what the study can contribute to the non-Colombian population, since otherwise the study would have no place in the international journal and otherwise its publication in a national journal would be more advisable.

Author Response

We have included a letter addressing the comments from all three reviewers, allowing for a comprehensive evaluation of their feedback.

Reviewer 2 Report

Thanks for inviting me to review this paper on knowledge, attitude and practice regarding bacterial resistance and antibiotic use in Medellin – Columbia.

I will first give some general comments, then some major and minor comments that follow the chronology of the paper.

General comments

I find the introduction too long and suggest focusing on what is common findings from other studies, and likewise on discrepancies from other studies. Are there any relevant geographical differences? What are the strengths and weaknesses by the methods used in these studies? Fig. 1 should be abandoned – the information content of the figure is covered by the text. A broader view on drivers for antibacterial resistance could be valuable, see e.g. A Chatterjee et al. Lancet Infect Dis 2018; 18: e368-78.

Major comments

Methods

Statistical testing & analysis

It appears that the authors have performed repeated statistical tests on the same data material without accounting for repeated test. This increases the probability of Type-1 errors.

The multiple linear regression (MLR) model is unclear to me. I suggest presenting a possible causal structure in a DAG-model (see www.dagitty.net). There is no information on whether the assumptions for the regression are met, and there is no information on regression diagnostics. The model has several categorical variables with more than two categories, and this would require dummy coding of the variable. However, this is not addressed in the text.

In addition, the outcome variable in MLR should ideally have no boundaries. As I read your model, the outcome is in the range 0-100. This might pose a problem if you have a high proportion of observations towards the extremes.

The respondents are from seven different areas in Medellin and the sample from each area represents the number of inhabitants. However, you do not use this information in the analysis. Is KAP homogeneous over the different areas?

Knowledge/Attitude/Practice index

Some statements contain more than one topic. This cause concern on what part of the statement the respondent is addressing. Some examples:

If antibiotics are consumed too constantly or misused….. Do the respondent answer on constant use or misuse? Or both?

Antibiotics are the same drugs used to relive pain and fewer. (Pain and fewer is obviously different)

Antibiotic resistance is a serious problem in other countries, not here in Colombia? (Are they addressing Colombia?)

Resistant infections only affect people in hospitals or with severe illness. (You can be hospitalized without receiving antibiotic treatment)

Similar examples can be found in the other indexes. I refrain from further comments on this.

The authors use a Likert scale without a neutral alternative. This choice should be addressed in the discussion.

Results

In Tab. 2, the authors present results from bivariate analysis. In my opinion, it would be beneficial to present the adjusted values in the same table.

See my comments above on the MLR.

Discussion

Focus on your own results in perspective of previous knowledge and avoid too much repetition of results and speculation. How do you know that there is a need for campaigns on consequences of non-prescription use? And impact on the city??

What is your most important finding? What is new?

Reliability of the instrument. In my opinion, some statements in the indexes would benefit from rephrasing to clarify what the respondents addresses.

A discussion on the statistical methods used is missing completely in the discussion.

Conclusion

The data do not support extrapolation to the entire Colombian population, and I am not confident that the methods used provide a reliable scale for measuring KAP in upcoming research.

Minor comments

Define age span for the age groups.

The language is of acceptable quality.

Author Response

(The authors gave the same response as above.)

Reviewer 3 Report

I would thank the authors for this interesting manuscript.

Even the manuscript is generally well presented I have some comments that could improve its quality.

1. Title: you can delete use since you begun with practice. I suggest to write it as:  Knowledge, attitude and practice regarding antibacterials  and their resistance in Medellín-Colombia: a cross-sectional study.

2. Delete  "[email protected]" from the first affiliation

3. Line 14: …., "elimination of duplicate items, validity assessment, 14 and reliability evaluation using Cronbach's alpha". Reformulate this sentence please (it is n incomplete sentence).

4. Add more keyword to the abstract

5. Line 38: add a reference after economic.

6. delete figure1. It has practically no sense.

7. Table 1: Add "variable" in the bottom of the table

 Separate between the variable please

You should chose what word to choose please (gender or sex) verify in all the manuscript please.

The same also for education and school education

8. Write and organize your titles according to the journal's template  please

9. For the knowledge, attitude,  and practice index I have two main remarks:

A – You should begin with the general results (the median score) and then you provide the details of each item.

B. You should make the results related to the determinants separately after the three titles. You can add a title : determinants (of factors associated with knowledge, attitude and practice explaining Table 2. (to avoid to repeat citing this table in each part).

10. In figure 1 and 2 : the horizontal axis should be 100 not 80 or 90.

11. Line 126: delete attitude at the end of the title (the same in Line 138)

12. Discussion: Justify the manuscript please

13. Line 155-157: Recently, various initiatives…drugs (add a reference please)

14. Line 176-177: …despite the absence of massive educational campaigns…response(15). There is a contradiction here.

15. Line 186: add a reference after bacteria. The same in line 190 (after bacterial resistance) and line 192 (after outbreaks).

16. Line 198: what allows you to  determine positive and negative attitude (what is your scale?).

17. Line 216: how did you qualify 63.9/100 as poor. For me it is not poor. What is your scale that allows you to say this?

18. Line 247: "Additionally….population". What do you mean here?.

19. Material and methods: Separate between the subtitles (study designs, population, sample size..) and the paragraphs please

20. Line 267-277: have you done really all these steps? How did you extract your questions? And how did you arrive to 32 questions used in your manuscript. Explain please

21. Line 290: "…, for attitudes was 0.706, and for practices was 0.731". correct the sentence please

22. Conclusion: The conclusion is very basic. Ty to develop with your main results please

23. the references (in the text and in the list) should be revised according to the journal's guidelines.

Minor editing of English language required

Author Response

(The authors gave the same response as above.)

Round 2

Reviewer 1 Report

Congratulations. The article has been substantially improved and is ready for its publication.

Reviewer 3 Report

I would thank the authors for their efforts to improve the quality of the manuscript. However, the manuscript is still lacking some rigor in the result's presentation and the authors have added some unecessary parts that reduced substantially the quality of the manuscript.

The main concern is related to the title: "Relationship between Knowledge, Attitudes, and Practices":

I did not undertand why did the authors modify this part, since the figure presented well their results. Also, making such a long table in the manuscript make it unreadible (this should be added in the supplementary materials). Additionnaly, this table is repeated in line 233 with another explanation making this part unreadible. I suggest to remit the last figure and improve its quality and also reorganize your ideas.

I have also other concerns that the authors did not unswer during the first round:

regarding the survey, I didn't undertand how did the authors choose their questions from the 242 manuscript reviewed. I do not think that this part (the search method ) is interesting in this part (it is not a systematic review). you can just add which are the references used to obtain the items. Otherwise, you should add exactely how did you obtain the items of your questionnaire.

somme errors are also observed:

line 10: delete "use" after antibiotic

line 24,25: gender should be in lower case

Line 53-61: "Studies .....". which studies ? are they the cited studies (in this cas you should write: "these studies..") or other studies (here you should add the references).

Line 188: delete attitude at the last of the title

Line 208-213: where is the explanation? you should explain your results

At last, the form of the manuscript needs to be reviewed. the paragraphs are not justified and the form did not meet the criteria of the journal

The authorst did not use the same font size in all parts especially for the title of the table 

The reference are not in agreement with MDPI style (closing barckets not parenthesis) 

 Minor editing of English language required

Round 3

Reviewer 3 Report

Thank you for adressing all my comments

.